# miR526b and miR655 Induce Oxidative Stress in Breast Cancer

**DOI:** 10.3390/ijms20164039

**Published:** 2019-08-19

**Authors:** Bonita Shin, Riley Feser, Braydon Nault, Stephanie Hunter, Sujit Maiti, Kingsley Chukwunonso Ugwuagbo, Mousumi Majumder

**Affiliations:** Department of Biology, Brandon University, 3rd Floor, John R. Brodie Science Centre, 270—18th Street, Brandon, MB R7A6A9, Canada

**Keywords:** MicroRNA (miRNA), miR526b, miR655, oxidative stress, reactive oxygen species (ROS), superoxide (SO), Thioredoxin Reductase 1 (TXNRD1), breast cancer

## Abstract

In eukaryotes, overproduction of reactive oxygen species (ROS) causes oxidative stress, which contributes to chronic inflammation and cancer. MicroRNAs (miRNAs) are small, endogenously produced RNAs that play a major role in cancer progression. We established that overexpression of miR526b/miR655 promotes aggressive breast cancer phenotypes. Here, we investigated the roles of miR526b/miR655 in oxidative stress in breast cancer using in vitro and in silico assays. miRNA-overexpression in MCF7 cells directly enhances ROS and superoxide (SO) production, detected with fluorescence assays. We found that cell-free conditioned media contain extracellular miR526b/miR655 and treatment with these miRNA-conditioned media causes overproduction of ROS/SO in MCF7 and primary cells (HUVECs). Thioredoxin Reductase 1 (TXNRD1) is an oxidoreductase that maintains ROS/SO concentration. Overexpression of *TXNRD1* is associated with breast cancer progression. We observed that miR526b/miR655 overexpression upregulates *TXNRD1* expression in MCF7 cells, and treatment with miRNA-conditioned media upregulates *TXNRD1* in both MCF7 and HUVECs. Bioinformatic analysis identifies two negative regulators of TXNRD1, TCF21 and PBRM1, as direct targets of miR526b/miR655. We validated that *TCF21* and *PBRM1* were significantly downregulated with miRNA upregulation, establishing a link between miR526b/miR655 and TXNRD1. Finally, treatments with oxidative stress inducers such as H_2_O_2_ or miRNA-conditioned media showed an upregulation of miR526b/miR655 expression in MCF7 cells, indicating that oxidative stress also induces miRNA overexpression. This study establishes the dynamic functions of miR526b/miR655 in oxidative stress induction in breast cancer.

## 1. Introduction

Breast cancer is the most common cancer affecting women and is responsible for the highest number of cancer-related deaths among women worldwide [1]. Breast cancer progression follows a complex multistep process, which depends on multiple exogenous and endogenous factors. The production of reactive oxygen species (ROS) such as superoxide (SO) leads to the induction of oxidative stress, which has been largely associated with breast cancer [2]. Oxidative stress is the result of cellular inability to neutralize and eliminate excess ROS, which is frequently associated with cancer development and progression. Under normal physiological conditions, cells endogenously produce ROS such as H_2_O_2_, ONOO^−^, OH^−^, HClO^−^, NO^−^, ROO^−^, and SO, during metabolism, respiration, and biosynthesis of macromolecules. Thus, cell metabolites are great resources for understanding oxidative stress. Excessive ROS production can induce inflammation, regulate the cell cycle, and stimulate intracellular transduction pathways, which leads to the promotion of cancer [3]. Specifically, SO production is the consequence of oxygen (O_2_) acting as the final electron acceptor in the electron transport chain, and has been shown to regulate signaling cascades that lead to cell survival and proliferation [4]. Within the cell, there is a homeostatic balance of various protective molecules and ROS. However, in cancer, tumor cells demonstrate deviations in oxidative metabolism and signaling pathways as a result of the constitutive activation of growth signaling pathways, leading to increased levels of ROS and induction of oxidative stress [5].

A high concentration of ROS is a signature feature of the tumor microenvironment. Cells have a natural defense mechanism to reduce damage caused by oxidative stress. Antioxidants, which are stable molecules that donate electrons to neutralize free radicals, belong to this natural defense mechanism of the cell [6]. Cellular detoxification pathways are regulated by enzymes that eliminate ROS, which include SO dismutase, catalase, glutathione peroxidase, cysteine, and thioredoxin (TXN). Specifically, TXN is a ubiquitous antioxidant protein that is responsible for the regulation of dithiol/disulfide balance [7,8]. TXN is active when it is in its reduced form. When active, it will participate in a reaction catalyzed by peroxiredoxin to neutralize H_2_O_2_ and peroxynitrate, both of which are products of oxidative stress activity [9]. TXNRD1 is responsible for the conversion of TXN into its active state (Figure 1). Malfunctions in antioxidant pathways can lead to increased oxidative stress and consequential damage to the cells. High expression of *TXNRD1* is associated with increased oxidative stress and correlates with poor prognosis in breast cancer [10]. In cancers, excessive production of ROS can cause mutations in the DNA, overexpression of tumor-promoting microRNAs (miRNAs, miRs), release of inflammatory molecules, and inactivation of oxidoreductive enzymes; making antioxidant pathways dysfunctional. Overexpression of oncogenic miRNAs leads to the regulation and promotion of tumor growth; however, the regulation of oxidative stress in cancer by miRNAs remains unclear.

miRNAs are small, endogenously produced RNAs which regulate gene expression at the post-transcriptional level [11]. Release of circulating miRNAs in the tumor microenvironment can regulate tumor growth and metastasis. Previously, miR526b and miR655 have been established as oncogenic and tumor-promoting miRNAs in human breast cancer [12,13,14]. The roles of miR526b and miR655 have been implicated in many hallmarks of cancer, including: Driving primary tumor growth, induction of stem-like cell (SLC) phenotypes, epithelial-to-mesenchymal transition (EMT), invasion and migration, distant metastasis. We have shown that cell metabolites and cell-free conditioned media of these two miRNA-high cells induce tumor-associated angiogenesis and lymphangiogenesis in breast cancer [15]. It has also been shown that cellular stress and ROS production can also induce oncogenic miRNA expression in tumors, and it is well-established that both ROS and miRNA expression signatures are associated with tumor development, progression, metastasis, and therapeutic response [16]. Thus, we wanted to investigate the relationship between ROS and miR526b/miR655 in breast cancer.

In this study, we investigate the roles of oncogenic miR526b and miR655 in oxidative stress in breast cancer. First, we show that both miR526b/miR655 directly and indirectly regulate oxidative stress. Next, we use the expression of *TXNRD1* as a molecular marker of oxidative stress to further validate the link between miRNA and ROS production. Moreover, we identify a positive feedback loop between oxidative stress and miRNA expression in breast cancer, showing that while the upregulation of miR526b and miR655 led to the induction of ROS production, the induction of oxidative stress also further upregulated miR526b and miR655 expression in breast tumor cells. Hence, we establish the dynamic roles of miR526b and miR655 in oxidative stress in breast cancer.

## 2. Results

To test the effects of miR526b and miR655 in oxidative stress in breast cancer, we used an estrogen receptor (ER)-positive, poorly metastatic breast cancer cell line, MCF7, and highly aggressive, miR526b/miR655-overexpressing MCF7-miR526b and MCF7-miR655 cell lines. We also used a primary endothelial cell line, human umbilical vein endothelial cells (HUVEC), to test the indirect or paracrine effects of miR526b and miR655 on oxidative stress induction. Finally, we used a breast epithelial cell line MCF10A and breast cancer cell lines T47D, MCF7, SKBR3, MCF7-COX2, Hs578T, and MDA-MB-231 to measure *TXNRD1* expression.

### 2.1. miR526b and miR655 Directly Induce Oxidative Stress by Overproduction of ROS and SO

#### 2.1.1. Fluorescence Microplate Assay

Previously, studies have used a total ROS detection kit for the measurement of ROS and SO in triple negative breast cancer cell lines, colon cancer cells, colorectal cancer cell lines, and in hepatocellular carcinoma cells [17,18,19,20,21]. We used the same ROS-ID Total ROS/SO detection kit (Enzo Life Sciences, Farmingdale, NY, USA) to measure fluorescence due to ROS/SO production following manufacturer’s protocol. Microplate readings were carried out at 1 and 21 h following Pyocyanin (ROS inducer) treatment and addition of non-fluorescent, cell-permeable ROS detection dyes. We monitored cellular morphology at various time points from 1–24 h after the addition of the ROS inducer in MCF7 cells (data not shown). With minimum dosage of ROS inducer, we observed oxidative stress in the cell within an hour, and after 21 h a decrease in cell viability was recorded due to the toxicity of the ROS inducer. Therefore, fluorescence was measured at two different timepoints; at 1 and 21 h. Fluorescence emissions were captured using two different filters to detect green (Fluorescein) and red (Rhodamine) emissions. ROS/SO production was calculated by subtracting the negative control emissions (basal emissions) from the test group emissions (with treatment) (Appendix A). Overall, we found that ROS and SO production was greater in miRNA-high cells compared to MCF7 cells. Specifically, ROS production was found to be statistically significant at 21 h for MCF7-miR655 (Figure 2A). Similarly, SO production was found to be significantly greater in the MCF7-miR655 cell line compared to MCF7 at both 1 h and 21 h. ROS and SO production was not statistically significant for MCF7-miR526b compared to MCF7 (Figure 2A,B).

#### 2.1.2. Fluorescence Microscopy Assay

Fluorescence microscopy assays were conducted to measure the difference in cellular fluorescence expression with individual fluorescent cell quantification, determining the fraction of cells producing ROS and SO. Using the green (Fluorescein) and red (Rhodamine) fluorescence filter sets, photos of the fluorescent cells were captured with an inverted fluorescence microscope 1 h after the detection dyes were added. We also captured bright field images of cells without using fluorescence filters to quantify total number of viable cells (Appendix A). Results show that wells containing MCF7-miR526b (Figure 3D,E) or MCF7-miR655 cell lines (Figure 3G,H) had more fluorescing cells than MCF7 (Figure 3A,B) under both red and green filters. Similarly, quantifications show significantly higher green (Figure 3J) and red (Figure 3K) cells in both MCF7-miR526b and MCF7-MCF7-miR655-high cells compared to MCF7 cells. Furthermore, we measured the ratio of cells positive for both ROS and SO production using merged channels and found that miRNA-high cell lines (Figure 3F,I) had a significantly higher ratio of fluorescing cells under both filters compared to MCF7 cells (Figure 3C,L).

### 2.2. Cell-Free Conditioned Media from miR526b/miR655-High Cells Indirectly Induce Production of ROS and SO

The tumor microenvironment is very heterogeneous, containing tumor cells, endothelial cells, macrophages, miRNAs, cell metabolites, inflammatory molecules, growth factors, and also ROS. In the following assays, we first tested the paracrine effect of miRNA in oxidative stress. To test the paracrine effect of miRNA, we used the cell-free conditioned media from miR526b/miR655-high cells as an ROS inducer using MCF7 (tumor model) and HUVEC (primary endothelial model) cell lines. Next, we quantified pri-miR526b and pri-miR655 in the conditioned media to investigate if the indirect induction of oxidative stress in breast cancer is due to the presence of miR526b and miR655 in the cell secretions, and to justify our use of conditioned media as an ROS inducer.

#### 2.2.1. Fluorescence Microplate Assay with MCF7 Cells

MCF7 cells were grown and then treated with basal media or cell-free conditioned media (containing cell metabolites and secretory proteins) collected from MCF7-miR526b and MCF7-miR655 cells for 24 h. Then we added the ROS inducer as described earlier and fluorescence data were collected at 1 and 21 h. These two time points were selected to remain consistent with our previous experiments that used the ROS inducer. MCF7 cells treated with miRNA-conditioned media showed significantly higher ROS production than the basal media treated MCF7 control group at both 1 and 21 h. Specifically, the change is extremely significant for MCF7 cells treated with MCF7-miR526b conditioned media at 1 h, and with MCF7-miR655 conditioned media at 21 h (Figure 4A). Similarly, MCF7 cells treated with MCF7-miR655 cell-free conditioned media had significantly higher SO production than the basal media treated cells at 1 h (Figure 4B). At 21 h, MCF7 cells treated with MCF7-miR526b cell-free conditioned media had a significantly higher SO production than MCF7 cells treated with basal media. While MCF7 cells treated with MCF7-miR655 cell-free conditioned media did show slightly higher SO production than MCF7 treated with basal media, this was not statistically significant (Figure 4B).

#### 2.2.2. Fluorescence Microplate Assay with HUVECs

Previously, we have shown that cell-free conditioned media from miRNA-high cells induce angiogenic potential in HUVECs [15]. Here, we tested if cell-free conditioned media containing all secretory proteins and metabolites from miR526b/miR655-high cells can induce oxidative stress in HUVECs. HUVECs treated with MCF7-miR526b or MCF7-miR655 conditioned media for 12–18 h had significantly higher ROS/SO production compared to HUVECs treated with basal media (Figure 4C,D). It should be noted that HUVECs are very sensitive to changes in growth conditions and treatments, as they can only survive for 12–18 h without native growth condition. Thus, HUVECs were treated with conditioned media from miRNA-high cells for 12 h. We found that HUVECs were extremely stressed, observing cell death after an hour following the addition of the ROS inducer (Appendix A). Therefore, the microplate assay was done only 30 min after ROS inducer was added.

#### 2.2.3. Fluorescence Microscopy Assay with MCF7 Cells in miRNA- Conditioned Media

In this experiment, cell-free conditioned media was used as an inducer of oxidative stress. MCF7 cells were grown and treated with basal media or cell-free conditioned media from MCF7-miR526b or MCF7-miR655 cells for 12–18 h. No other ROS inducer was added, only cell-permeable dyes from the ROS detection kit were added to detect cell-free conditioned media-induced oxidative stress. Images were captured after 1 h, and the number of fluorescent cells were measured with ImageJ as mentioned above. Results show that MCF7 cells treated with MCF7-miR526b (Figure 5D–F) or MCF7-miR655 conditioned media (Figure 5G–I) had more fluorescing cells than basal media treated MCF7 cells (Figure 5A–C) for both Fluorescein and Rhodamine filters. Quantification of MCF7 cells treated with MCF7-miR526b or MCF7-miR655 conditioned media show a significant increase in ROS production (Figure 5J) and SO production (Figure 5K). The ratio of cells positive for both ROS and SO production was also significantly higher in cells treated with MCF7-miR526b or MCF7-miR655 conditioned media than those treated with basal media (Figure 5L).

#### 2.2.4. miRNA-High Cells Release miR526b and miR655 in Cell-Free Conditioned Media

To test if the indirect induction of oxidative stress with conditioned media is due to the presence of miRNA itself, we measured pri-miR526b and pri-miR655 expression in MCF7, MCF7-miR526, and MCF7-miR655 cell-free conditioned media. We found that both pri-miRNAs’ expressions were significantly higher in MCF7-miR526b conditioned media compared to MCF7 conditioned media (Figure 6). The expression of pri-miR526b was significantly higher and the expression of pri-miR655 was marginally higher in MCF7-miR655 conditioned media. It should be noted that in the MCF7-miR526b conditioned media, the overall expression of pri-miR526b was higher than pri-miR655, while in MCF7-miR655 conditioned media, the overall expression of pri-miR655 was higher than pri-miR526b (Figure 6). This result confirms that due to the release of miRNA in the conditioned media of serum starved cells, extracellular miR526b and miR655 act as an ROS inducer, therefore indirectly inducing oxidative stress in nearby cells.

### 2.3. TXNRD1 is a Marker for Oxidative Stress

TXN is an antioxidant protein that is responsible for neutralizing ROS within the cell [8]. TXNRD1 is the enzyme responsible for reducing TXN into its active form. Previous analyses of *TXNRD1* expression have shown that *TXNRD1* is upregulated in pancreatic, colon, lung, prostate, and breast cancers, and is associated with poor cancer prognosis [10]. To further investigate the direct and indirect roles of miR526b and miR655 in the induction of oxidative stress, *TXNRD1* was validated as a marker of oxidative stress using various breast cancer cell lines and its expression was measured in miR526b/miR655-high cell lines. Furthermore, bioinformatic analysis was done to investigate the regulation of *TXNRD1* by miR526b and miR655, which showed that miR526b and miR655 target two transcription factors that regulate *TXNRD1* expression. The expression of these transcription factors was then measured in miR526b/miR655-high cell lines. Moreover, with the success of using miRNA-conditioned media as an ROS inducer in our previous assays, we tested to see if cell-free conditioned media from miR526b/miR655-high cells regulate *TXNRD1* expression in both tumor and endothelial cells.

#### 2.3.1. Highly Metastatic Breast Cancer Cell Lines Show Upregulation of TXNRD1

MCF10A, T47D, MCF7, SKBR3, MCF7-COX2, Hs578T, and MDA-MB-231 cell lines were used to quantify the expression of *TXNRD1* using qRT-PCR. Since MCF10A is a breast epithelial cell line, gene expression changes for all breast cancer cell lines were measured and compared to MCF10A. Results show that *TXNRD1* was significantly downregulated in the poorly metastatic MCF7 and SKBR3 cell lines, while the T47D cell line showed no change in expression (Figure 7A). *TXNRD1* was significantly upregulated in all highly metastatic cell lines, MCF7-COX2, Hs578T, and MDA-MB-231; with maximum upregulation seen in MDA-MB-231 (Figure 7A). We have previously found that these aggressive breast cancer cell lines (MCF7-COX2, Hs578T, and MDA-MB-231) show overexpression of both miR526b and miR655; while poorly metastatic cells (MCF7, T47D) show low expression of both miRNAs [12,13]. These observations validate the use of *TXNRD1* as a marker of oxidative stress in breast cancer, and show a link between *TXNRD1*, miR526b, and miR655 expression.

#### 2.3.2. miRNA Overexpression Directly Upregulates TXNRD1 Expression

To establish the direct role of miRNA in oxidative stress, total RNA extraction followed by qRT-PCR was carried out with MCF7, MCF7-miR526b, and MCF7-miR655 cell lines to quantify the expression of *TXNRD1.* Results show that *TXNRD1* was significantly upregulated in both MCF7-miR526b and MCF7-miR655 cell lines compared to MCF7, with greater fold change in *TXNRD1* expression measured in the MCF7-miR655 cell line (Figure 7B).

#### 2.3.3. Bioinformatic Analysis to Identify a Link between miRNAs and TXNRD1

Since we observed that miRNA overexpression results in the upregulation of *TXNRD1* in breast cancer, we further wanted to investigate this mechanism in silico. Thus, we conducted bioinformatic analysis to investigate how miR526b and miR655 regulate *TXNRD1* expression. Both miRNA target gene lists were extracted from the miRBase database, using TargetScan analysis tool which can predict miRNA target genes in mammalian mRNA pool [22,23,24,25,26]. By virtue, miRNAs bind to target genes, degrading the corresponding mRNA at the post-transcriptional level, and thus block the protein expression of the target. We found that *TXNRD1* is not a direct target of miR526b and miR655, so we instead attempted to identify transcription factors (TFs) that regulate *TXNRD1* and are also targets of miR526b and miR655. In miR526b/miR655 overexpressing cells, we observed that *TXNRD1* expression is high, which indicates that these miRNAs might be targeting negative regulators of *TXNRD1*. To identify these TFs, we used Enrichr, a tool that consists of both a validated user-submitted gene list and a search engine for further analysis [27]. By comparing miRNA target genes and *TXNRD1* regulatory TFs, we identified eight TFs as direct targets of miR526b (blue down arrows in the yellow circle) and eleven TFs as direct targets of miR655 (blue down arrows in the pink circle) (Figure 8A). Finally, we identified two TFs (PBRM1 and TCF21) as common targets of both miRNAs (blue down arrows in the green circle), which negatively regulate *TXNRD1* (Figure 8A). Both *PBRM1* and *TCF21* have been shown to have tumor suppressor-like functions in breast cancer [28,29]. Therefore, we hypothesize that when miR526b and miR655 are upregulated, their targets *PBRM1* and *TCF21* are downregulated, leading to the upregulation of *TXNRD1*. This result justifies the abundance of *TXNRD1* in miRNA-high cells.

#### 2.3.4. miRNA Overexpression Indirectly Upregulates TXNRD1 by Targeting Negative Regulator of the Gene

The expression of *PBRM1* and *TCF21* was measured in MCF7, MCF7-miR526b, and MCF7-miR655 cell lines to further confirm that miR526b and miR655 target these TFs to regulate the expression of *TXNRD1*. Results show that both *PBRM1* and *TCF21* are significantly downregulated in both miR526b/miR655-high cell lines as compared to MCF7 cells, validating the in silico analysis (Figure 8B).

#### 2.3.5. MCF7 Cells Treated with miR526b and miR655-High Cell-Free Conditioned Media Show Upregulation of TXNRD1

MCF7 cells were treated with basal media or miR526b/miR655-high cell-free conditioned media for 12–18 h as mentioned before. RNA extraction and gene expression assays were carried out to quantify the expression of *TXNRD1*. It was found that *TXNRD1* expression in MCF7-miR655 conditioned media-treated cells was significantly higher compared to the basal media treated MCF7 cells. MCF7 cells treated with MCF7-miR526b conditioned media had marginally higher, but statistically non-significant *TXNRD1* expression compared to basal media treated MCF7 cells (Figure 9A). These results piqued our interest in the paracrine effect of miRNA-overexpressing cells. In the tumor microenvironment, oxidative stress in neighboring normal, immune, and endothelial cells would also be increased. Thus, we wanted to investigate this principle in non-cancerous cells, using a primary endothelial cell line (HUVECs).

#### 2.3.6. HUVECs Treated with Cell-Free miR526b and miR655 Conditioned Media Show Upregulation of TXNRD1

HUVECs were treated with basal media, MCF7-miR526b conditioned media, or MCF7-miR655 conditioned media for 12 h. Using qRT-PCR, quantification for the expression of *TXNRD1* in treated and non-treated HUVECs was performed. Results show that HUVECs treated with MCF7-miR526b conditioned media containing secretory proteins and metabolites show a marginal upregulation of *TXNRD1* compared to HUVECs treated with basal media, but was not statistically significant (Figure 9B). However, HUVECs treated with MCF7-miR655 conditioned media show a significant overexpression of *TXNRD1* when compared to HUVECs treated with basal media (Figure 9B).

### 2.4. Cell-Free miRNA Conditioned Media Indirectly Induces miRNA Overexpression in MCF7 Cells

Since we have shown that cell-free conditioned media from miR526b/miR655-high cell lines induces ROS production and *TXNRD1* expression in MCF7 cells, we wanted to test if cell-metabolites and secretory proteins could also induce oncogenic miRNA upregulation in poorly metastatic MCF7 cells. MCF7 cells were treated with serum-free basal media, MCF7-miR526b conditioned media, or MCF7-miR655 conditioned media for 21 h. RNA was extracted and reverse transcribed into cDNA to quantify pri-miR526b and pri-miR655 expressions. After relative gene expression analysis, the results showed that all miRNA conditioned media-treated MCF7 cells had a significant increase in expression of pri-miR526b and pri-miR655 compared to basal control MCF7 cells (Figure 9C,D). These results establish the dynamic roles of miR526b/miR655 and ROS in the tumor microenvironment where a complex interplay between the tumor cell, tumor cell secretions, and endothelial cells is ongoing, thus promoting tumor growth.

### 2.5. Induction of Oxidative Stress Upregulates miR526b and miR655 Expression in MCF7 Cells

Finally, we wanted to investigate if miR526b and miR655 are key responders to oxidative stress. Thus, we induced oxidative stress in MCF7 cells using a chemical inducer, H_2_O_2_, and measured its effects on miR526b and miR655 expression. MCF7 cells were grown until 80% confluent, and then treated with either_,_ 25 µM or 50 µM of H_2_O_2_ for 24 h. Following treatment, RNA was extracted and reverse transcribed into cDNA. qRT-PCR was then carried out to quantify the expression of pri-miR526b and pri-miR655 in the H_2_O_2_-treated and non-treated MCF7 cells. Results show a significant dose-dependent increase in the expression of pri-miR655 in H_2_O_2_-treated MCF7 cells at both 25 µM and 50 µM concentrations (Figure 10). Expression of pri-miR526b following treatment with 50 µM of H_2_O_2_ showed marginal upregulation; however, this was not statistically significant (Figure 10). These results support the notion that these two miRNAs are immediate responders to oxidative stress in breast cancer.

## 3. Discussion

Previously, we have established the roles of oncogenic miR526b and miR655 in breast cancer disease progression, angiogenesis, cancer stem cell regulation, and metastasis [12,13,14,15]. We have also previously shown that overexpression of miR526b and miR655 is associated with poor breast cancer patient survival and found that miRNA expression was elevated in advanced grades of breast cancer [12,13], suggesting these two miRNAs are oncogenic and metastasis-promoting miRNAs. Here, we tested the potential roles of these miRNAs in the induction of oxidative stress and the effects of this potential regulation within the tumor microenvironment. ROS including SO, free radicals, and charged ions are the byproducts of cellular metabolism. Under normal physiological conditions, cells keep a balance of ROS production and neutralization to maintain tissue homeostasis [4,5]. However, overproduction of ROS induces oxidative stress, which is associated with cancer development and progression. Production of ROS causes DNA mutation, oncogenic miRNA expression, protein malfunction, apoptosis, and the induction of oxidative stress, which has been identified as a major cause of breast cancer [30]. Superoxide (SO) serves as a growth-stimulating molecule that regulates signaling cascades, which leads to cell survival and proliferation [4]. Moreover, it has been shown that ER-positive breast cancer tumor samples exhibit higher SO levels compared to matched normal tissues, and that SO levels are higher in the blood of breast cancer patients [31,32]. In this study we used the ER-positive MCF7 breast cancer cell line as an in vitro tumor model to establish the link between miRNA and ROS/SO production in breast cancer. The link between various miRNAs and oxidative stress has also been previously reported, such as the expression of miR155 shown to regulate oxidative stress in endothelial cells, and the in vitro induction of oxidative stress being shown to regulate the expression of miR146a and miR34a [33,34]. Oxidative stress is the result of excess ROS, which is due to an imbalance between the generation of ROS and the cell’s ability to neutralize and eliminate them. Previous studies have linked the roles of miRNA with ROS production; for example, Zhang et al. showed that miR21 modulates oxidative stress by measuring ROS production in cells through ROS detection [35].

We wanted to investigate if miRNA expression can regulate ROS production and induce oxidative stress, while oxidative stress can also regulate miRNA expression in breast cancer. In this study, we used cell-permeable dyes which interact with cellular ROS and SO to detect and quantify ROS/SO production in cells using fluorescence assays. First, we measured and compared ROS/SO production in MCF7, MCF7-miR526b, and MCF7-miR655 cell lines, to test for the direct regulation of oxidative stress in breast cancer cells by these miRNAs. We have shown that MCF7-miR526b and MCF7-miR655 cell lines, especially the MCF7-miR655 cell line, have higher production of ROS/SO than MCF7 cells, showing that miR526b and miR655 have a role in the endogenous or “direct” induction of oxidative stress.

Breast tumors consist of heterogeneous cells and interactions between tumor cells and cells within the tumor microenvironment to promote tumor sustenance and metastasis. Specifically, cell metabolites and secretions from tumor cells into the tumor microenvironment function to communicate between tumor cells with nearby non-tumor cells, which can regulate many different pathways and networks to promote tumor metastasis [5,36]. Therefore, we investigated the paracrine or “indirect” induction of oxidative stress by treating MCF7 cells and HUVECs with tumor cell metabolites and secretions from miR526b/miR655-high cell lines. We observed that cell-free conditioned media collected from miR526b/miR655-high cells induced ROS/SO production in both MCF7 cells and HUVECs, which suggests that miR526b and miR655 secretory proteins and metabolites indirectly induce oxidative stress in the tumor microenvironment.

The roles of extracellular or cell-free miR526b and miR655 in the complexity of breast tumor metastasis has not been well investigated. Although in recent years many reports studied the detection of miRNAs in the blood of cancer patients, it was only recently shown by a group that extracellular miRNAs can be found in the media of *Drosophila* cell lines growth in petri dish [37]; giving an excellent model to test cell-free miRNA in vitro. We previously have shown that miR526b/miR655 cell-free conditioned media contain stimulatory proteins which induce angiogenesis in the tumor microenvironment [15]. However, we never measured the presence of miR526b/miR655 themselves in the cell-free conditioned media. Here, for the first time, we showed that cell-free supernatant (cell-free conditioned media) collected from MCF7, MCF7-miR526b, and MCF7-miR655 serum-starved cells media contain miR5256b/miR655. Moreover, we found that both MCF7-miR526b and MCF7-miR655 conditioned media had a higher expression of both pri-miR526b and pri-miR655 compared to the MCF7 conditioned media. These results prove that cell secretions from miR526b/miR655-high cell lines also contain miRNAs and indirectly play a role in oxidative stress induction in the tumor microenvironment.

Since ROS activates signaling cascades that promote cell survival and tumor growth, it is expected that highly metastatic and aggressive breast cancer cell lines will be under higher oxidative stress than poorly metastatic breast cancer cell lines [38]. Here, we observed that a key regulatory protein of oxidative stress, TXNRD1, is upregulated in highly metastatic and aggressive breast cancer cell lines, which is supported by other studies showing a link between oxidative stress and breast cancer [10,39]. Next, it was found that miRNA overexpression induced *TXNRD1* expression in MCF7-miR526b and MCF7-miR655 cell lines. These results led us to investigate potential targets of miR526b and miR655 to explain the upregulation of *TXNRD1* in miRNA-overexpressing cell lines. It was found that two transcription factors, PBRM1 (polybromo 1) and TCF21 (Transcription Factor 21), which are negative regulators of *TXNRD1*, are both targets of miR526b and miR655. *PBRM1* has been described as a tumor suppressor gene that is responsible for the control of the cell cycle [28]. Low *PBRM1* expression has been shown to predict poor prognosis in breast cancer and mutations in *PBRM1* have been reported in many tumor types such as renal cell carcinoma, biliary carcinoma, gallbladder carcinoma, and intrahepatic cholangiocarcinoma [28,40]. *TCF21* has also been reported as a tumor suppressor gene in gastric cancer, colorectal cancer, head and neck carcinomas, and breast cancer [29,41,42,43]. Following the Bioinformatics analysis, we validated this observation by measuring the expression of these two TFs in MCF7, MCF7-miR526b, and MCF7-miR655 cell lines. Our results showed that *PBRM1* and *TCF21* are indeed downregulated in miR526b/miR655-high cell lines, proving that miR526b and miR655 upregulate *TXNRD1* by targeting these two negative regulators of *TXNRD1*.

In the tumor microenvironment, dynamics between tumor cell secretion of inflammatory molecules and growth factors, communication with endothelial cells, and activation of immune cells are well established [15,44]. We have previously shown that treatment of HUVECs with MCF7-miR526b or MCF7-miR655 conditioned media induced cancer related phenotypes, such as angiogenesis and lymphangiogenesis via paracrine regulation [15]. In addition, here we showed that even cell-free conditioned media contain miR526b/miR655. To further investigate the roles of miR526b and miR655 in the indirect induction of oxidative stress, *TXNRD1* expression was quantified and compared in MCF7 cells and HUVECs treated with MCF7-miR526b or MCF7-miR655 cell-free conditioned media. Here, we have found that in both MCF7 and HUVECs treated with MCF7-miR526b or MCF7-miR655 cell-conditioned media, there is an upregulation of *TXNRD1*, which supports our findings of miR526b and miR655 indirectly regulating the production of ROS and induction of oxidative stress.

While ROS production is a component of the cell’s physiological process, high concentrations of ROS are detrimental for the cell, which induces apoptosis. However, epigenetic changes, such as miRNA overexpression by tumor cells, protect cellular death and promote cell proliferation. It has been shown that the induction of oxidative stress can alter the expression of specific miRNAs by inhibiting or inducing their expression [16]. Similarly, we have shown that conditioned media collected from MCF7-miR526b and MCF7-miR655 cell lines induce ROS production in MCF7 cells, thus miR526b and miR655 are involved in the regulation of oxidative stress both directly, and indirectly.

Next, we tested to see if miR526b and miR655 are immediate responders to cellular oxidative stress. In this study, we have shown that cell-free conditioned media collected from MCF7-miR526b and MCF7-miR655 cell lines induce oxidative stress; thus, we again used conditioned media from miRNA-overexpressing cells to induce oxidative stress in MCF7 cells and examined miR526b and miR655 expression. We observed that MCF7 cells treated with MCF7-miR526b or MCF7-miR655 conditioned media had increased expressions of both pri-miR526b and pri-miR655. Interestingly, we observed that MCF7 cells treated with miR526b conditioned media and metabolites showed a higher expression of pri-miR526b than pri-miR655, and MCF7 cells treated with conditioned media from MCF7-miR655 showed a higher expression of pri-miR655 than pri-miR526b. It has previously been shown by other groups that H_2_O_2_ treatment induces oxidative stress in MCF7 cells [45,46]. Therefore, to further validate that miR526b and miR655 are immediate responders to cellular oxidative stress, we tested the effects of H_2_O_2_ treatment on MCF7 cells. Interestingly, H_2_O_2_ treatment significantly increased the expression of pri-miR655 in MCF7 cells, and marginally increased pri-miR526b expression in a dose dependent manner. Taken together, these results suggest that a positive feedback loop exists between oxidative stress and miRNA in breast cancer, which is driven by miRNA-high cell line secretions.

Interestingly, we noticed a common trend in which miR655 appeared to have a stronger role in both the direct and indirect induction of oxidative stress than miR526b. MCF7-miR655 was shown to have the greatest expression of *TXNRD1*, and the greatest production of ROS/SO as compared to MCF7 cells. Furthermore, MCF7 and HUVECs treated with cell-free conditioned media from MCF7-miR655 showed the greatest expression of *TXNRD1* and the greatest amount of ROS/SO production. This shows that while miR526b still appeared to be involved in oxidative stress and the *TXNRD1* pathway, miR655 has a stronger role in oxidative stress pathways in breast cancer. Differential roles of miRNAs in regulating oxidative stress may be due to various targets of miRNAs (Figure 8A). In the future, it would be interesting to investigate the signaling pathways involved in miR526b and miR655′s regulation of oxidative stress in breast cancer.

In this study, we identified the novel roles of miR526b and miR655 in oxidative stress in breast cancer. Specifically, this is the first time that miR526b and miR655 has been linked to oxidative stress, as we show that miR526b and miR655 regulate ROS production, as well as show greater expression of miRNAs during cellular oxidative stress. Furthermore, we suggest a positive feedback loop exists between miR526b/miR655 and oxidative stress in breast cancer. Here we also show that miR526b and miR655 are present in the extracellular tumor microenvironment, which suggests that these cell free miRNAs might also be regulating extracellular signaling and regulating oxidative stress hence promoting tumor growth and metastasis. These discoveries add to the accumulation of evidence that miR526b and miR655 are strong candidates for potential biomarkers in breast cancer. Future studies require a complete analysis of miRNA cell metabolites and cell secretome to discover new functions of miR526b and miR655. This will allow us to discover complex mechanisms behind oxidative stress induction in breast cancer and the possibility of these miRNAs as therapeutic targets to abrogate oxidative stress.

## 4. Materials and Methods

We conducted all experiments at Brandon University, following the regulations of Brandon University Research Ethics (#21986, approved on April 21, 2017) and Biohazard Committee (#2017-BIO-02, approved on September 13, 2017). An overview of the methods workflow is presented in Figure 11.

### 4.1. Cell Culture

All human breast cancer cell lines MCF7, SKBR3, T47D, MDAMB231, and Hs578T were purchased from American Type Culture Collection (ATCC, Rockville, MD, USA). All breast cancer cell lines were grown in Dulbecco’s Modified Eagle Medium (DMEM) (Gibco, Mississauga, ON, Canada) supplemented with 10% fetal bovine serum (FBS) and 1% Penstrep as described before following manufacturer protocols [12,13,47]. Stable miRNA-overexpressing MCF7-miR526b and MCF7-miR655 cell lines were established as previously described [12,13]. MCF7-miR526b and MCF7-miR655 cells were grown in Roswell Park Memorial Institute (RPMI) 1640 medium (Gibco, Mississauga, ON, Canada) supplemented with 10% FBS and 1% Penstrep. Furthermore, MCF7-miR526b and MCF7-miR655 cell lines were sustained with Geneticin (Gibco, Mississauga, ON, CAN) at 40 mg/mL. An immortalized non-tumorigenic mammary epithelial cell line MCF10A was cultured and maintained by Ling Liu at the University of Western Ontario in Professor Peeyush K Lala’s laboratory as described earlier [47] and they kindly shared an aliquot of MCF10A cDNA.

HUVECs were purchased from Life Technologies (NY, USA) and grown in Medium 200 (Gibco, Mississauga, ON, Canada), supplemented with Low Serum Growth Supplement Kit containing 2% FBS, hydrocortisone (1 µg/mL), human epidermal growth factor (10ng/mL), basic fibroblast growth factor (3 ng/mL), and heparin (10 µg/mL). All cell lines were maintained in a humidified incubator at 37 °C with 5% CO_2_.

### 4.2. Collection of Conditioned Media

MCF7, MCF7-miR526b, and MCF7-miR655 cell lines were grown in complete RPMI 1640 until 90% confluent. Cells were then washed with phosphate buffered saline (PBS) to remove any trace of the complete media. The cells were then starved with basal RPMI 1640 medium (serum-free) for 12–16 h prior to collection of media, and then centrifuged. Cell-free supernatant was then collected for assays testing the indirect induction of oxidative stress by miR526b and miR655. We hypothesized that these cell supernatants contain cell metabolites and secretory proteins with unknown function.

### 4.3. RNA Extraction and Quantitative Real-Time PCR

Total RNA was extracted from all cell lines using the miRNeasy Mini Kit (Qiagen, Toronto, ON, Canada) and reverse transcribed using the microRNA and mRNA cDNA Reverse Transcription Kit (Applied Biosystems, Waltham, MA, USA). For conditioned media miRNA extraction, MCF7, MCF7-miR526b, and MCF7-miR655 conditioned media were centrifuged at 3000 RPM for 5 min, and the supernatants were collected for RNA extraction following the miRNeasy Mini Kit protocol (Qiagen, Toronto, ON, Canada). The TaqMan miRNA or Gene Expression Assays was used for qRT-PCR. The expressions of two endogenous control genes, *Beta-actin* (Hs01060665_g1) and *RPL5* (Hs03044958_g1), were quantified using qRT-PCR and were used to normalize the expression of *TXNRD1* (Hs00917067), *PBRM1* (Hs01015916_m1), *TCF21* (Hs00162646_m1), pri-miR526b (Hs03296227), and pri-miR655 (Hs03304873) markers using relative analysis. Gene expression was measured using CT values from each curve, which are obtained from the point at which each curve reaches the threshold. To determine the relative levels of gene expression, the comparative threshold cycle method (Δ*C*t) was used [15,47].

### 4.4. Fluorescence Microplate Assay

MCF7, MCF7-miR526b, and MCF7-miR655 cells were seeded in a 96-well plate as shown in Appendix A and were grown until 70% confluent. Total ROS and SO levels were detected using the ROS-ID Total ROS/SO detection kit (Enzo Life Sciences, Farmingdale, NY, USA) according to manufacturer’s instructions. Negative controls and test groups were prepared for each cell line. The negative controls were treated with 5 μM of *N*-acetyl-l-cysteine (ROS inhibitor) for 30 min to eliminate all ROS present in the cells. Following this, 200 μM of Pyocyanin (ROS inducer) was added to induce ROS production in all wells. The test groups were treated with only the ROS inducer. Detection reagents from the ROS-ID kit were used to measure ROS/SO production. Microplate readings were done at 1 and 21 h following the addition of detection dyes, using the standard Fluorescein filter (Ex/Em: 485/535 nm) and Rhodamine filter (Ex/Em: 550/625 nm). Data was collected using the SoftMax Pro 6 Microplate Data Acquisition and Analysis software (Molecular Devices, San Jose, CA, USA). Concentrations of the ROS inhibitor, inducer, and detection reagents were determined based on a known standard curve. For normalization, negative control emissions were subtracted from the test group emissions to show the total production ROS in each cell line (Appendix A).

Two more plate-reading experiments were done using MCF7 (Appendix A) or HUVECs (Appendix A) treated with basal media (no serum added) or MCF7-miR526b/miR655 conditioned media. MCF7 cells/HUVECs were seeded as shown in Appendix A, and when 70% confluent, they were washed with PBS to remove traces of the serum and growth factors. They were then treated with basal media or MCF7-miR526b/miR655 supernatant for 12–18 h. The assay was then performed as described above.

### 4.5. Fluorescence Microscopy Assay

We used the same ROS/SO detection kit to determine the number of cells producing ROS and SO following the manufacturer’s protocol. Test groups and negative controls were prepared for the MCF7, MCF7-miR526b, and MCF7-miR655 cell lines and seeded as described above. When 70% confluent, the cells were washed PBS and treated as described above. The assay was performed on the NIS Elements Advanced Research software (Nikon, Melville, NY, USA), using a Nikon Ds-Ri1 microscopy camera. The fluorescent cells in each experiment were quantified using the ImageJ software (National Institute of Health, Bethesda, MD, USA). Fluorescent images were converted to 8-bit and adjustments were made. Particle analysis was then done on ImageJ to quantify the number of fluorescing cells (Appendix A) and (Appendix A) For each condition, the negative control was used as a threshold for quantification (Appendix A). Negative control quantifications were subtracted from test group quantifications, and then divided by the total number of cells to present the total ROS/SO production in each cell line as ratios.

A second experiment was conducted using the same ROS-ID kit, following the same protocol as described above. MCF7 cells were seeded in a 96-well plate as shown in Appendix A, and once they have reached 70% confluency, washed with PBS and treated with miRNA-conditioned media. The assay was performed and fluorescent cells were quantified and presented using the same methods outlined above. The assay was then performed as described above.

### 4.6. Bioinformatics Analysis

A total of 4133 target transcript genes for human miR526b (hsa-miR-526b) and 3264 target transcription genes for human miR655 (hsa-miR-655) were found using TargetScan (analysis tool which can predict miRNA target genes in mammalian mRNA pool) and miRBase database [22,23,24,25,26]. Finding the TFs of the *TXNRD1* gene allows us to distinguish what up/down regulates *TXNRD1* expression within the human system.

We used the Enrichr (a tool that consists of both a validated user-submitted gene list and a search engine for further analysis) and found 155 TFs perturbations followed by gene expression [27]. These 155 TFs upregulate or downregulate *TXNRD1* gene expression. We then compared the two data sets to find common genes between miR526b/miR655 target genes and *TXNRD1* regulatory genes (TFs). We observed 19 genes that are common between miR526b targets and *TXNRD1* regulators, of which 8 down-regulated the *TXNRD1* gene expression. We then compared the gene list to find common genes between miR655 targets and *TXNRD1* regulators, and observed 18 genes that are common in both gene sets, of which 11 down-regulated the *TXNRD1* gene. Finally, we found two TFs as common targets of both miRNAs that are negative regulators of *TXNRD1.* To determine the target a nominal *p* < 0.05 was used and the p value was calculated with Fisher exact test, which is a proportion test that assumes a binomial distribution and independence for the probability of any gene belonging to any set.

### 4.7. Treatment of MCF7 Cells with H_2_O_2_

MCF7 cells were grown and maintained until 90% confluent. H2O2 at a concentration of either 25 μM or 50 μM was added to confluent MCF7 cells for 24 h. H_2_O_2_ was used instead of pyocyanin to test the effects of a different ROS inducer. These concentrations of H_2_O_2_ have been previously reported to induce oxidative stress in the MCF7 cell line [45,46]. Following the addition of H_2_O_2_ for 24 h, MCF7 cells were collected for RNA extraction, carried out with miRNeasy Mini Kit (Qiagen, Toronto, ON, Canada) and reverse transcribed using the TaqMan microRNA and mRNA cDNA Reverse Transcription Kit (Applied Biosystems, Waltham, MA, USA). qRT-PCR was carried out as mentioned above to measure pri-miR526b and pri-miR655 expression, and were normalized to *Beta-actin* and *RPL5.*

### 4.8. Statistical Analysis

Statistical calculations were performed using GraphPad Prism software version 8 (https://www.graphpad.com/quickcalcs/ttest1/?Format = SEM). All parametric data were analyzed with one-way ANOVA followed by Tukey–Kramer or Dunnett post-hoc comparisons. Student’s t-test was used when comparing two datasets. Statistically relevant differences between means were accepted at *p* < 0.05. Fisher exact test was performed for miRNA database and target TFs analysis followed by false positive rate (FDR) correction to identify significant changes in target gene expression (*p* < 0.05).

## Figures and Tables

**Figure 1 ijms-20-04039-f001:**
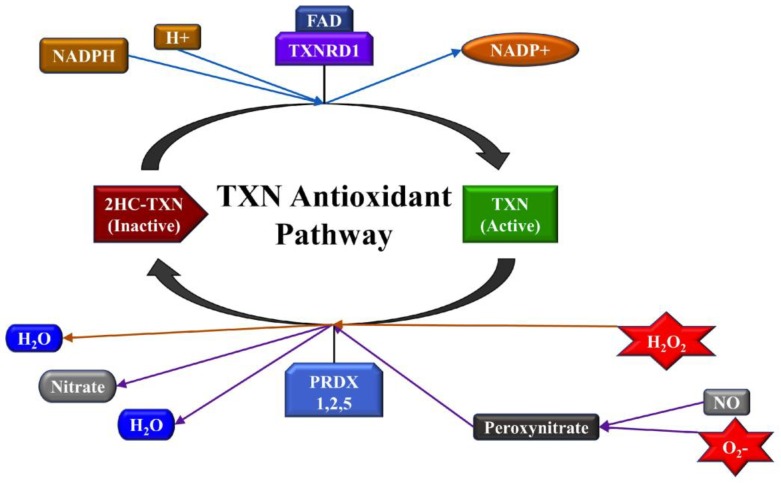
Thioredoxin (TXN) is a main constituent in an antioxidant pathway that neutralizes Hydrogen Peroxide (H_2_O_2_) and superoxide (O_2_−), to prevent oxidative damage. TXN exists in active (reduced) and inactive (oxidized) states. Thioredoxin Reductase 1 (TXNRD1) is responsible for reducing 2HC-TXN (TXN with attached double hydrocarbon) into its active form. Therefore, in the presence of more ROS, an increased expression of *TXNRD1* occurs to protect the cells from oxidative damage.

**Figure 2 ijms-20-04039-f002:**
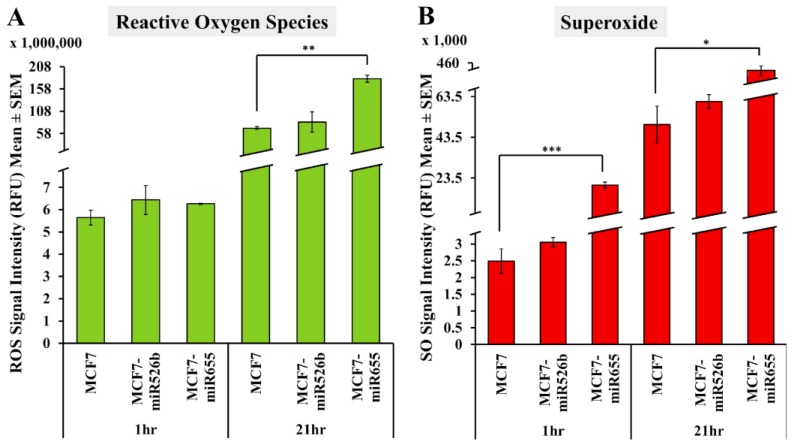
Fluorescence microplate assays to quantify ROS (Green) and SO (Red) production by MCF7, MCF7-miR526b, and MCF7-miR655 cell lines. (**A**) Quantitative data represents the ROS signal intensity in MCF7, MCF7-miR526b, and MCF7-miR655 cell lines at 1 and 21 h. (**B**) Quantitative data represents SO signal intensity in MCF7, MCF7-miR526b, and MCF7-miR655 cell lines at the 1 and 21 h. Data presented as the mean ± SEM of triplicate replicates; * *p* < 0.05, ** *p* < 0.01, *** *p* < 0.001.

**Figure 3 ijms-20-04039-f003:**
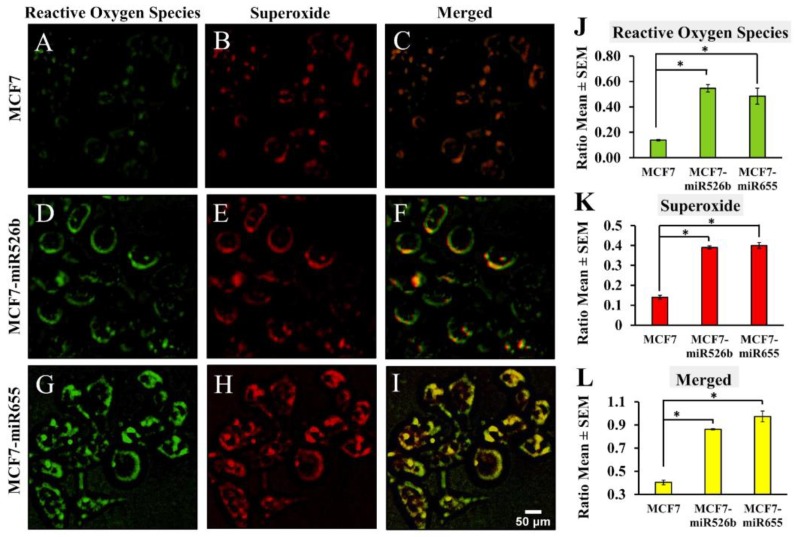
Fluorescence microscopy images and quantification of ROS/SO production in MCF7, MCF7-miR526b, and MCF7-miR655 cell lines. (**A**–**C**) Images of fluorescent MCF7 cells with green, red, or merged filters. (**D**–**F**) Images of fluorescent MCF7-miR526b cells with green, red, or merged filters. (**G**–**I**) Images of fluorescent MCF7-miR655 cells with green, red, or merged filters. Scale bar: 50 μm. (**J**) Quantification of ratios of cells positive for ROS detection. (**K**) Quantification ratios of cells positive for SO detection. (**L**) Quantification ratios of cells showing both ROS and SO production. Quantitative data presented as the mean ± SEM of triplicate replicates. Quantifications presented in ratios of fluorescence-positive cells to the total number of cells; * *p* < 0.05.

**Figure 4 ijms-20-04039-f004:**
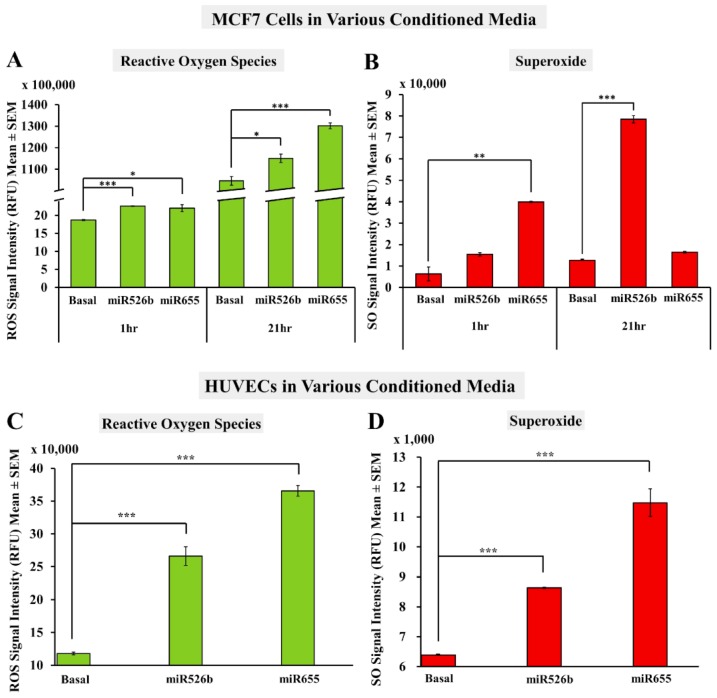
Fluorescence microplate assay with MCF7 and HUVEC cells cultured in miRNA conditioned media. (**A**) MCF7 cells treated with MCF7-miR526b or MCF7-miR655 conditioned media show an overproduction of ROS as compared to basal media treated cells at both 1 and 21 h. (**B**) MCF7 cells treated with MCF7-miR655 conditioned media show a significant overproduction of SO at 1 h, and MCF7 cells treated with MCF7-miR526b conditioned media show a significant overproduction of SO at 21 h compared to MCF7 cells treated with basal media. (**C**) HUVECs treated with MCF7-miR526b or MCF7-miR655 conditioned media show overproduction of ROS compared to HUVECs treated with basal media after 30 min. (**D**) HUVECs treated with MCF7-miR526b or MCF7-miR655 conditioned media show a significant overproduction of SO as compared to non-treated MCF7 cells after 30 min. Data presented as the mean ± SEM of triplicate replicates; * *p* < 0.05, ** *p* < 0.01, *** *p* < 0.001.

**Figure 5 ijms-20-04039-f005:**
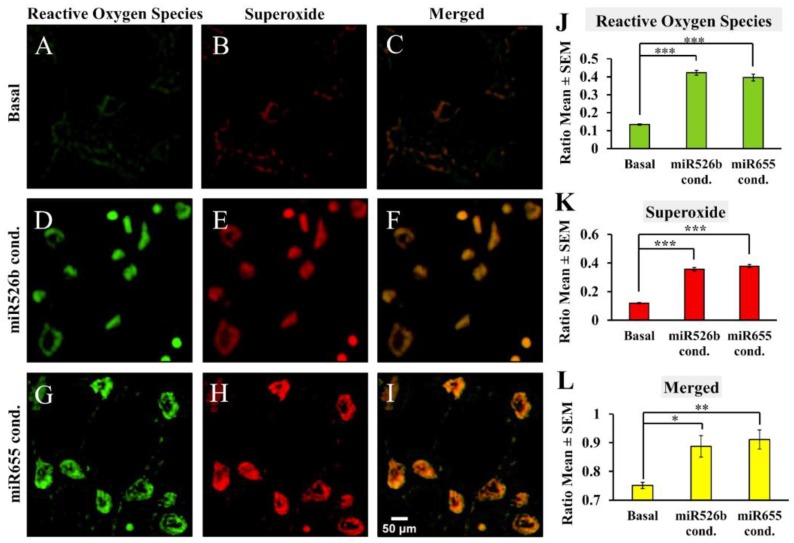
Fluorescence microscopy with MCF7 cell line treated with basal, MCF7-miR526b, or MCF7-miR655 cell-free conditioned media to quantify ROS/SO producing cells. MCF7 treated with basal media under the Rhodamine filter was used as a threshold to quantify ROS positive cells. (**A**–**C**) Images of MCF7 cells treated with basal media in green, red, or merged filters. (**D**–**F**) Images of MCF7 cells treated with cell-free conditioned media from MCF7-miR526b cells in green, red, or merged filters. (**G**–**I**) Images of MCF7 cells treated with cell-free conditioned media from MCF7-miR655 cells in green, red, or merged filters. Scale bar: 50 μm. (**J**) Quantification of cells positive for ROS detection presented as ratios. (**K**) Quantification of cells positive for SO detection presented as ratios. (**L**) Ratio of cells showing both ROS and SO production. Quantitative data presented as the mean ± SEM of quadruplicate replicates; * *p* < 0.05, ** *p* < 0.01, *** *p* < 0.001.

**Figure 6 ijms-20-04039-f006:**
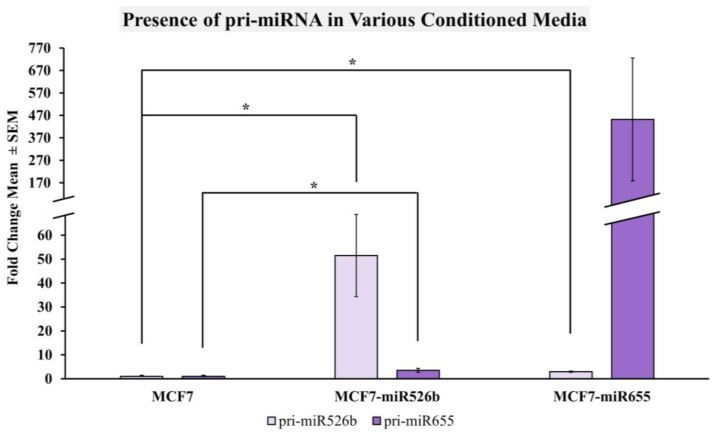
Expression of pri-miR526b and pri-miR655 in various conditioned media measured using qRT-PCR. MCF7-miR526b conditioned media show a significantly higher expression of both pri-miRNAs with prominent change in pri-miR526b expression compared to MCF7 conditioned media. MCF7-miR655 conditioned media show a significantly higher expression of miR526b, and very high expression of pri-miR655, which was not significant. Data is presented as the mean ± SEM of duplicate replicates; * *p* < 0.05.

**Figure 7 ijms-20-04039-f007:**
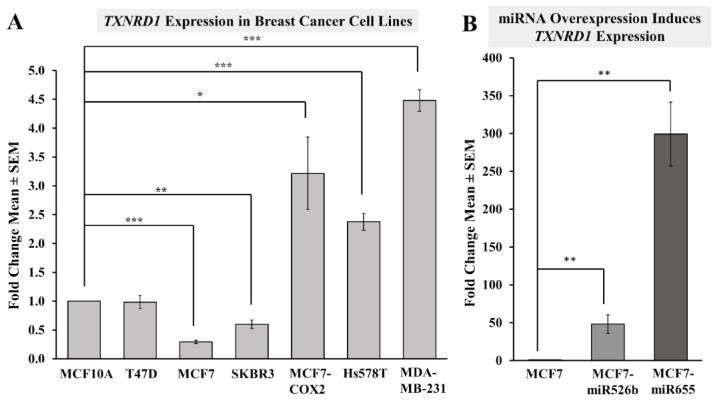
Expression of oxidative stress marker *TXNRD1* in various cell lines measured using qRT-PCR. (**A**) Breast cancer cell lines with various degrees of metastatic potential show a difference in *TXNRD1* expression. The more metastatic cell lines including MCF7-COX2, Hs578T, and MDA-MB-231 show the greatest fold change of *TXNRD1* expression and MCF7 cells showing lowest *TXNRD1* expression compared to the breast epithelial MCF10A cell line. (**B**) Expression of *TXNRD1* is quantified in MCF7 cells, MCF7-miR526b, and MCF7-miR655 cell lines, showing how these oncogenic miRNAs impact the expression of this oxidative stress marker. Large fold change increases are seen in both miRNA cell lines. Data is presented as the mean ± SEM of triplicate replicates; * *p* < 0.05, ** *p* < 0.01, *** *p* < 0.001.

**Figure 8 ijms-20-04039-f008:**
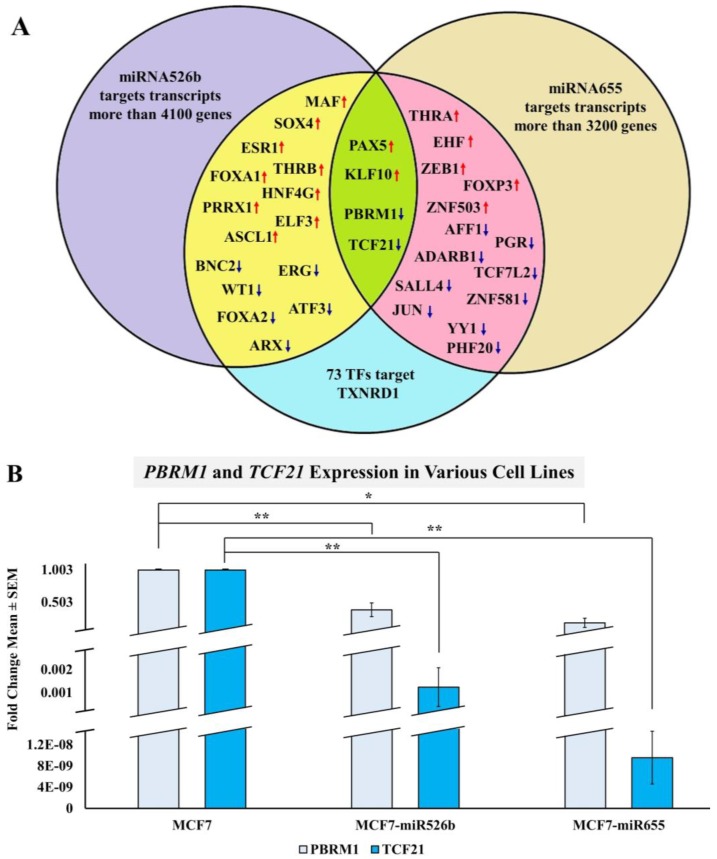
(**A**) Overlap of TFs regulating *TXNRD1*, and miR526b, miR655 target genes. The purple area represents the list of 4133 miR526b target genes, the brown area represents the list of 3264 miR655 target genes, and the blue area represents all 155 TFs regulating *TXNRD1*. The yellow area shows the miR526b target genes which are also TFs regulating *TXNRD1*. The pink area indicates miR655 targets which are also TFs regulating *TXNRD1*. The green center represents the overlap of all three criteria, which shows the four TFs of *TXNRD1* which are common targets of both miRNAs. The red up arrow symbolizes that the TF upregulates *TXNRD1* expression and the blue down arrow signifies that the TFs downregulates *TXNRD1* expression. Because we observed that *TXNRD1* is upregulated in miRNA-high cells, we considered both miRNAs targeting the two TFs, *PBRM1*, and *TCF21*, which are the negative regulators of *TXNRD1*. (**B**) *PBRM1* and *TCF21* expression in MCF7, MCF7-miR526b, and MCF7-miR655 cell lines. miR526b/miR655-high cell lines show significantly lower expression of both *PBRM1* and *TCF21*. This indicates both miRNAs target the negative regulator of *TXNRD1.* Data presented as the mean of quadruplicate replicates; * *p* < 0.01, ** *p* < 0.001.

**Figure 9 ijms-20-04039-f009:**
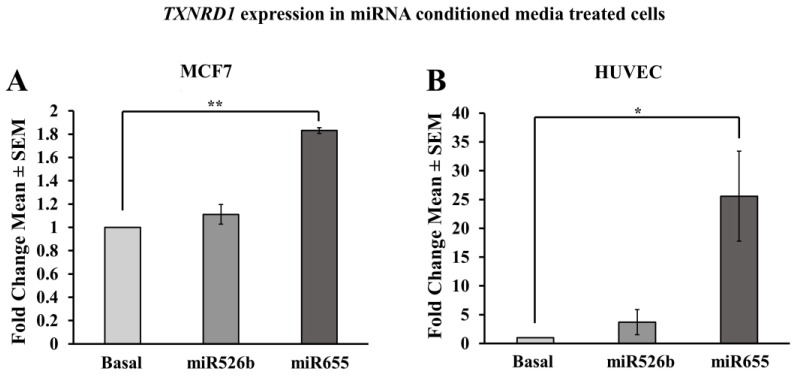
Indirect effects of miRNA overexpression on *TXNRD1,* pri-miR526b, and pri-miR655 expression. (**A**) *TXNRD1* expression in MCF7 cells treated with MCF7-miR526b or MCF7-miR655 conditioned media compared to non-treated MCF7 cells. (**B**) *TXNRD1* expression in HUVEC cells treated with MCF7-miR526b or MCF7-miR655 conditioned media compared to non-treated MCF7 cells. (**C**) pri-miR526b expression in MCF7 cells treated with MCF7-miR526b or MCF7-miR655 conditioned media compared to non-treated MCF7 cells. (**D**) pri-miR655 expression in MCF7 cells treated with MCF7-miR526b or MCF7-miR655 conditioned media compared to non-treated MCF7 cells. Data presented as the mean ± SEM of triplicate replicates; * *p* < 0.05, ** *p* < 0.001.

**Figure 10 ijms-20-04039-f010:**
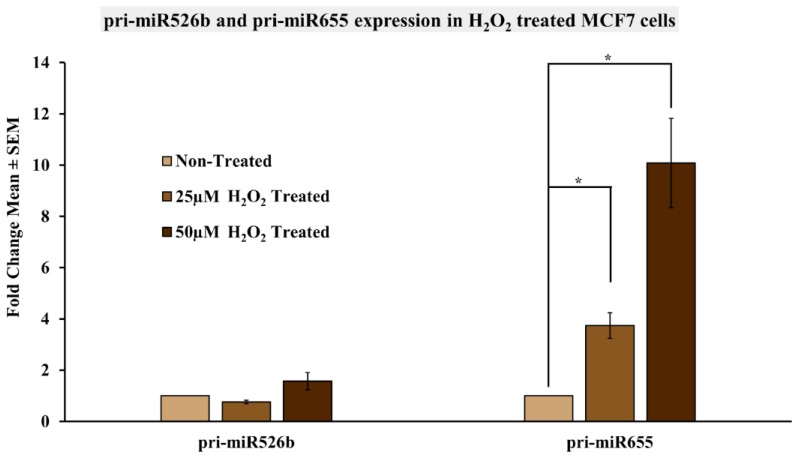
Expression of miR526b and miR655 in H_2_O_2_ treated MCF7 cells. pri-miR526b and pri-miR655 expression quantified in MCF7 cells, MCF7 cells treated with 25 µM H_2_O_2_, or 50 µM H_2_O_2_ using qRT-PCR. Data presented as the mean ± SEM of triplicate replicates; * *p* < 0.05.

**Figure 11 ijms-20-04039-f011:**
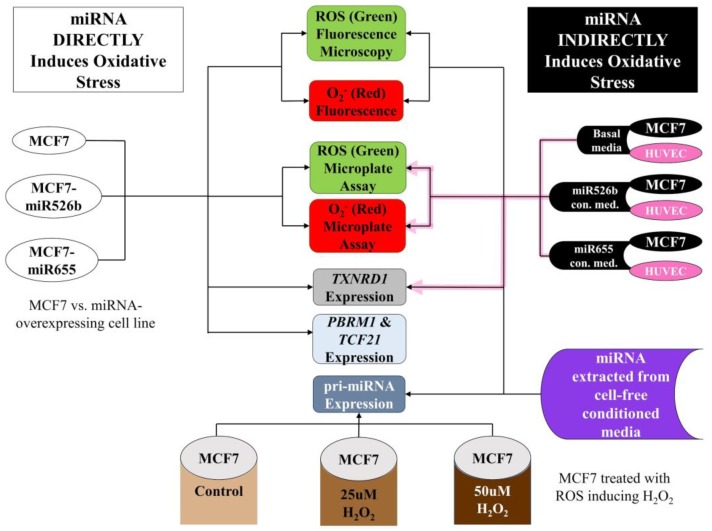
Outline of the in vitro approaches taken in establishing the direct and indirect induction of oxidative stress by miR526b and miR655, as well as the effect of ROS induction on the regulation of miRNA.

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
