# Peer review of "miR526b and miR655 Induce Oxidative Stress in Breast Cancer"

_ijms, 2019, doi:10.3390/ijms20164039_

Round 1

Reviewer 1 Report

The authors presented a nice work to investigate the role of miR-526b and miR-655 in regulating oxidative stress. They observed that miRNA-overexpression in MCF7 cells directly increases ROS and superoxide (SO) production. they proposed a small circuit among two miRNAs and direct (PBRM1 and TCF21) and indirect targets (TXNRD1) using in vitro and in silico assays.  

- pag. 9  miRNA target gene lists were extracted using TargetScan and miRBase databases. the authors took the union or intersection between the two lists derived from the two different databases? please specify

- Could the authors give more detail about Enrich database? How the database find transcription factor? is it a validated or predicted database?

- in the figure 7 the authors seem  to find 73 TFs but in the main text the authors focused on 2 TFs. It is not clear how the authors passed from 73 TFs to 2 TFs. please clarify better

- It is not clear.  TXNRD1 is  not a primary target of two miRNAs. is it right? is it  confirmed from miRNA-target database?

- no analyses are presented for primary miRNA target PBRM1 and TCF21. maybe it could be interesting to see if the two mRNAs are down-regulated when the miRNAs are over-expressed. actually the proposed circuit miRNA-mRNA is only speculated.

Reviewer 2 Report

Title: miR526b and miR655 Induce Oxidative Stress in Breast Cancer

In this manuscript, the authors investigated the role of miR526b and miR655 on regulation of ROS in breast cancer cell MCF-7 and HUVECs. The direct and indirect effect of miR526b and miR655 on TXNRD1 have been determined by experimental evidences. In addition, the potential regulatory mechanism between miR526b and miR655 on TXNRD1 have been evaluated via bioinformatic databases. This study provides novel insights miR526b and miR655 in oxidative stress in breast cancer; however, more experiments are essential for improving this manuscript.

1.      Why were Fluorescence microplate assays performed at 1 and 21 hours following Pyocyanin (ROS inducer) treatment? Please add rationale.

2.      The correlation between miR526b and miR655 expression, and TXNRD1 was observed in several breast cancer cell lines. Is the expression of miR526b and miR655 correlated with clinical outcome via online database? Such as metastatic-free survival?

3.      The effect of MCF7-miR526b or MCF7-miR655 conditioned media was investigated in this study. How could the author confirm miR526b and miR655 presented in conditioned media? The presence of these extracellular miRNAs should be confirmed by additional experiments.

4.      The bioinformatic analysis found two potential TXNRD1 regulators, PRRM1 and TCF21. This is an interesting finding. Therefore, the expression of PBRM1 and TCF21 should be measured in MCF-7 and MCF7-miR526b or MCF7-miR655 cells. It could further confirm the possible regulatory mechanism between miRNAs and PBRM1 and TCF21 via experimental evidences.

Round 2

Reviewer 1 Report

The manuscript has been significantly improved

Reviewer 2 Report

The manuscript is acceptable to me and I have no further comments.